# Multi-Modal Biological Destruction by Cold Atmospheric Plasma: Capability and Mechanism

**DOI:** 10.3390/biomedicines9091259

**Published:** 2021-09-18

**Authors:** Dayun Yan, Alisa Malyavko, Qihui Wang, Kostya (Ken) Ostrikov, Jonathan H. Sherman, Michael Keidar

**Affiliations:** 1Department of Mechanical and Aerospace Engineering, George Washington University, Washington, DC 20052, USA; qwang52@gwmail.gwu.edu; 2School of Medicine and Health Science, George Washington University, Washington, DC 20052, USA; alisamalyavko@gwmail.gwu.edu; 3School of Chemistry and Physics, Queensland University of Technology (QUT), Brisbane, QLD 4000, Australia; kostya.ostrikov@qut.edu.au; 4Centre for Biomedical Technologies, Queensland University of Technology (QUT), Brisbane, QLD 4000, Australia; 5WVU Medicine-Berkeley Medical Center, West Virginia University, Martinsburg, WV 25401, USA; jsherman0620@gmail.com

**Keywords:** cold atmospheric plasma, cell death, virus inactivation, cancer therapy, microorganism sterilization

## Abstract

Cold atmospheric plasma (CAP) is a near-room-temperature, partially ionized gas composed of reactive neutral and charged species. CAP also generates physical factors, including ultraviolet (UV) radiation and thermal and electromagnetic (EM) effects. Studies over the past decade demonstrated that CAP could effectively induce death in a wide range of cell types, from mammalian to bacterial cells. Viruses can also be inactivated by a CAP treatment. The CAP-triggered cell-death types mainly include apoptosis, necrosis, and autophagy-associated cell death. Cell death and virus inactivation triggered by CAP are the foundation of the emerging medical applications of CAP, including cancer therapy, sterilization, and wound healing. Here, we systematically analyze the entire picture of multi-modal biological destruction by CAP treatment and their underlying mechanisms based on the latest discoveries particularly the physical effects on cancer cells.

## 1. CAP and Plasma Medicine

CAP is a near-room-temperature ionized gas composed of products including neutral particles, such as neutral atoms and molecules; charged particles, such as ions; electrons; and diverse, long-lived and short-lived reactive species, such as reactive oxygen species (ROS) and reactive nitrogen species (RNS) [1,2,3,4]. CAP is also referred to as nonthermal plasma (NTP), cold plasma, physical plasma, and gas plasma in many references [5,6,7,8,9]. CAP is a non-equilibrium plasma in which heavy particles have effective temperatures close to room temperature through weak elastic collisions during the discharge process [10]. CAP also generates several physical effects, including thermal effect, UV effect, and EM effect [11,12].

Three types of CAP sources have been widely used in plasma medicine and can be roughly divided into three categories: direct discharge sources, indirect discharge sources, and hybrid discharge sources (Figure 1) [13]. In the direct discharge source, such as volume dielectric barrier discharge (DBD), sample (animals/tissues/cells) is one of the electrodes participating in the discharge. In the indirect discharge source, such as CAP jet, the formed plasma will be transported by a gas (such as Helium) flow from the main discharge arc area to affect the samples, which do not participate in the discharge. Finally, in the hybrid discharge source, such as surface DBD, the grounded, mesh-shaped electrode generates plasma within the mesh and allows nearly all current to flow through mesh rather than samples (such as skin). Despite different morphologies, power input, and reactive species generation in CAP, their chemical composition and physical effects are quite similar. CAP can be precisely controlled by modulating basic operational parameters (gas flow rate, etc.) and discharge parameters (such as discharge voltage, current, duty circle, etc.) [14,15].

Plasma generated by these sources can be used to directly touch biological samples, which exposes samples to reactive species and physical factors (Figure 2a). Alternatively, biological adaptive solutions, such as medium and phosphate buffered saline (PBS), can be used as a carrier of these long-lived reactive species to exert a killing effect on viruses and cells [16,17]. Reactive species will cause oxidative stress in the CAP-treated cells particularly cancer cells and, finally, trigger cell-death pathways if the reactive species’ dose on a single cell is sufficiently large [18,19]. The biological effects of physical factors have been hypothesized for a long time but without clear evidence until very recently. One possible candidate is the EM effect from CAP jet, which causes a structural damage on melanoma cells and triggers a quick necrosis [20].

Multimodal chemical and physical nature of CAP makes it a suitable, controllable, flexible, and even a self-adaptive tool for many medical applications, ranging from microorganism sterilization, dermatitis, wound healing, and cancer therapy [21,22,23,24,25,26,27,28]. For microorganism sterilization, particularly bacterial inactivation, CAP can cause strong damage on both gram-positive and gram-negative bacteria, including some multi-drug resistant bacteria (Figure 2b) [29,30,31]. The biofilm composed of complex microorganism community can also be effectively inactivated by CAP treatment (Figure 2c) [32,33,34,35]. These anti-bacterial capacities of CAP may be the foundation to drastically improve wound healing efficacy (Figure 2d) [29,36,37,38,39]. Over the past decade, CAP has shown impressive potential as a novel anti-cancer tool both in vitro and in vivo. CAP can selectively kill many cancer cell lines while having only limited side effects on normal counteracting cell lines [40,41,42,43]. Importantly, a simple CAP treatment just on the skin above the subcutaneous tumor site could effectively decrease the tumor size and extend life in mice, which demonstrates the non-invasive potential of CAP as a novel anti-cancer modality (Figure 2e) [8,44,45,46]. Besides, the inactivation of viruses by CAP has also been reported in many studies and has recently been summarized [47]. For example, the strongly inhibited infectivity of feline calicivirus on its host kidney cells has been demonstrated (Figure 2f) [48]. The CAP-triggered cell death or virus inactivation is the foundation of nearly all these applications.

**Figure 2 biomedicines-09-01259-f002:**
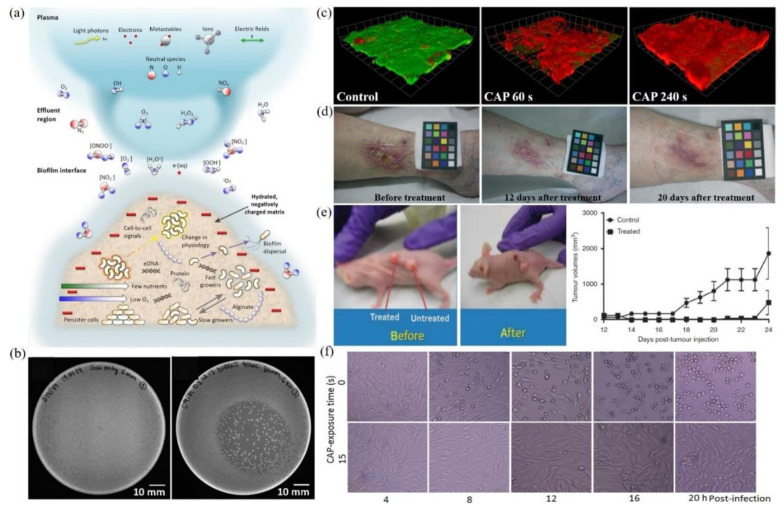
A general picture of plasma medicine. (**a**) Schematic illustration of the biological effect of CAP. Here, biofilm was used as an example [26]. (**b**) Growth inhibition of CAP source (DBD) on bacteria (S. Typhimurium) grown on LB agar plates: control (left) and DBD treatment (right) [31]. (**c**) Confocal scanning laser microscopic imaging of strong bacterial death in a CAP-treated biofilm grown on polycarbonate coupons. Live cells and dead cells were present in green and red, respectively [30]. (**d**) The improved wound healing of the inflamed ulcer on a human leg by CAP treatment [39]. (**e**) Tumor inhibition effect of CAP jet treatment on subcutaneously xenografted bladder tumor on mice [8]. (**f**) The inactivation of feline calicivirus and the inhibited infectivity on its host Crandell-Reese feline kidney cells by CAP treatment [48].

## 2. Apoptosis

Apoptosis is a specific form of programmed cell death, which plays an important role in a large number of human pathologies, including neurodegenerative disorders, autoimmune diseases, and cancer [49,50]. A typical apoptosis in mammalian cells is characterized by cell shrinkage, cytoplasmic membrane blebbing, chromatin condensation, nuclear membrane breakdown, as well as the final formation of vesicles from plasma membrane knowns as apoptotic bodies [51]. The formed apoptotic bodies have complete membranes and will be rapidly cleaned by phagocytes, which will not trigger inflammatory response in vivo [51,52].

The basic features and mechanism of apoptosis in mammalian cells have been extensively investigated. Generally, for the extrinsic apoptosis pathway, apoptosis is triggered by the activation of death receptors, such as tumor necrosis factor receptor (TNFR), and the further recruitment of death-inducing signaling complex (DISC), which cleavages caspase-8 zymogen and further activates effector caspase-3/7 [53,54]. In contrast, the intrinsic apoptosis pathway is based on the mitochondria-mediated and the death receptor-mediated apoptotic events [55]. Death stimulus, such as radiation; cytokine deprivation, like growth factor; cytotoxic drugs, like reactive species, and the DNA damage due to UV irradiation or oxidative stress, will initiate a series of pro-apoptotic events, including the phosphorylation of p53 (p-p53), the further activated expression of B-cell lymphoma 2 (BCL-2) homology region 3 (BH3)-only proteins, such as BH3 interacting-domain death agonist (Bid), BCL-2 associated agonist of cell-death protein (BAD), BCL2 and adenovirus EB1 19 kDa-interacting protein 3-like (NIX), p53 upregulated modulator of apoptosis (PUMA), and Phorbol-12-myristate-13-acetate-induced protein 1 (NOXA) [52,56,57,58]. These pro-apoptotic factors form BCL-2-associated X protein (BAX)/BCL-2 homologous antagonist killer (BAK) channels on mitochondrial membranes to realize mitochondrial outer membrane permeabilization (MOMP), which facilitates the release of apoptosis-inducing proteins, such as cytochrome c (cyt c) and other intermembrane mitochondrial components, such as second mitochondria-derived activator of caspase (Smac), HtrA serine peptidase 2 (Omi), Endonuclease G (Endo G), and apoptosis-inducing factor (AIF), into the cytosol [49,51,59,60]. The inhibited function of BCL-2 family on mitochondrial membrane also facilitates the release of mitochondrial proteins [61]. In cytosol, cyt c binds to apoptotic protease activating factor-1 (Apaf-1) and forms apoptosome, which further recruits procaspase-9 to form apoptosome-caspase-9 holoenzyme [62,63]. The activated caspase-9 further activates caspase-3/7 zymogen to be functional effector caspase-3/7 and begins a series of apoptotic events [59].

Typical apoptotic events include nucleus and DNA fragmentation by caspase-activated DNase (CAD), cytoskeleton breakdown, cellular membrane budding, and final apoptotic bodies. Apoptosis will not cause the release of cytosol components into extracellular environment. The dying cells in vivo will be rapidly scavenged by phagocytes, such as macrophages and dendritic cells [64]. In the last stage, apoptotic cells will produce “find me” signals, such as lysophosphatidylcholine signal, and “eat me” signals, such as the externalization of phosphatidylserine on cellular membrane [65,66,67]. The phosphatidylserine on plasma membrane will be bound by proteins such as Annexin, which will be recognized and bound by phagocytic cells [68]. The fluorescein-labeled Annexin V has been widely used as a biomarker in many apoptosis detection methods, including fluorescent imaging and flow cytometry [69].

For mammalian cells, apoptosis is the most widely observed cellular response to CAP treatment, which can produce a sufficiently high concentration of reactive species in aqueous solution, such as medium. Apoptosis has been widely observed in the CAP-treated cancer cells [41,70,71]. Apoptosis is triggered by CAP-originated reactive species particularly ROS [72,73,74,75]. Both extracellular and intracellular ROS scavengers, such as N-Acetyl-cysteine (NAC), and apoptosis inhibitors, such as zVAD, can effectively inhibit apoptosis in the CAP-treated mammalian cells [76,77,78,79,80,81,82]. The characterization of apoptosis is currently mainly based on flow cytometric data [83,84,85,86,87], accompanied with the analysis based on terminal deoxynucleotidyl transferase dUTP nick end labeling (TUNEL) assay, western blot, polymerase chain reaction (PCR), caspase 3/7 assay, mitochondrial membrane potential assay, cyt c assay, p53 assay, and BAX assay [41,71,79,88,89,90,91].

Based on these studies, the CAP-triggered apoptosis strictly follows the well understood apoptotic events and pathways. The dominant CAP-triggered cancer cells’ death followed the caspase-dependent apoptosis pathways, though a few studies reported the caspase-independent apoptosis in the CAP-treated cancer cells [73,92]. The release of cyt c into cytosol, the expression of p-p53/p73/p38/c-Jun N-terminal kinases (JNK), NOXA, Bax, BCL-2, caspase-8, cleavage of caspase-9, caspase-3/7, the cleavage of poly (ADP-ribose) polymerase (PARP), the loss of mitochondrial transmembrane potential, and DNA fragmentation have been widely observed in the CAP-treated cancer cells [77,78,81,83,93,94,95,96,97,98,99,100]. In contrast, the direct microscopic imaging on the apoptotic process, particularly the key features of apoptosis, such as budding of cells and the formation of apoptotic body, was largely lacking in most studies. A typical observation of apoptotic CAP-treated melanoma cells is shown in Figure 3a. The chronological expression of key proteins followed the well-known apoptotic pathways. Activation of p53 is the key step to trigger apoptosis. A caspase/apoptosis-independent cell death was reported for the CAP-treated p53-mutated glioblastoma multiforme cells. CAP induced rapid cell death by the accumulation of lysosomes without triggering autophagy [101]. The lack of TP53 gene expression might inhibit the caspase-dependent apoptotic pathways.

The significantly modified redox balance in the CAP-treated cancer cells has been regarded as the main reason to trigger apoptosis. The significant intracellular ROS rise has been widely observed in dozens of cell lines [46,76,91,102,103]. The rise of ROS caused a great deal of damage to organelles and important molecules, such as mitochondria, endoplasmic reticulum (ER), lysosome [104], DNA, cellular membrane, and extracellular matrix [76,77,86,87,94,97,105,106,107]. All these damages might further trigger apoptosis. Among them, DNA damage has been extensively investigated. The main DNA damage type was the double-strand break (DSB), though another type, like the oxidation on bases, such as 8-hydroxyl-2′-deoxyguanosine (8-OHdG), has also been reported [70,108]. The expression of γ-H2AX has been widely observed shortly after CAP treatment [80,93,109,110,111,112]. The serine 139 on H2AX is phosphorylated by ataxia telangiectasia mutated (ATM) recruited on DSB site with other DNA damage-response complexes [113]. The enhanced expression of ATM has been observed in the CAP-treated oral cavity squamous cell carcinoma cells and melanoma cells [112,114]. The increased level of p-p53 in the CAP-treated mouse melanoma cells B16F10 just occurred after the expression of γ-H2AX, which strictly followed the typical chronological order of DNA damage response (Figure 3b,c).

## 3. Autophagy-Associated Death

The autophagy-associated cell death has been proposed as a novel mode of cell death very different from apoptosis [115]. Autophagy was first found as a survival mechanism when cells, such as yeasts, were under sublethal stress, such as nutrient deprivation or lacking growth factors in extracellular environment [60,115,116]. The autophagic cells survive under these stresses by digesting their own organelles and macromolecules to recycle their own nonessential or damaged organelles or macromolecular components [115,117]. The cells that do not receive nutrients for extended periods of time will ultimately digest all available substrates and die, which is a autophagy-associated cell death [115,118]. Autophagic cell death is characterized by the presence of abundant autophagosomes in dying cells [119]. Autophagy can also lead to cell death under oxidative stress conditions, such as a ROS exposure [120].

Three forms of autophagy have been defined on the basis of how lysosomes receive material for degradation: macroautophagy, microautophagy, and chaperone-mediated autophagy [115]. Macroautophagy degrades cytosolic material via sequestration into double-membrane vesicles called autophagosomes that subsequently fuse with lysosomes [116]. Here, we focus on macroautophagy. The general molecular mechanism of autophagy has been systematically introduced in previous summaries [60,121,122,123,124]. Here, we give a brief introduction of typical nonspecific macroautophagy pathway. The well-known autophagy process can be divided into five stages: initiation, vesicle nucleation, vesicle elongation, vesicle fusion, and cargo degradation [121]. Three types of vesicles are formed chronologically in this process: phagophore, autophagosome, and autophagolysosome.

Autophagy is initiated by upstream activation through various conditions of stress, such as starvation, hypoxia, oxidative stress, protein aggregation, ER stress, and others [121,123]. In starvation case, the decrease in glucose transport releases the inhibition effect of mammalian target of rapamycin (mTOR) complex 1 (mTORC1) on UNC-51-like kinase 1 (ULK1) complex (ULK1, ULK2, FAK family kinase-interacting protein of 200 kDa (FIP200), autophagy-related protein (ATG)101, and ATG13.) [60,121]. Oxidative stress, such as ROS stress, inhibits phosphoinositide 3-kinase (PI3K) pathway but also inhibits the function of mTOR, which facilitates the initiation of autophagy [120]. The core autophagic pathway starts from the formation of an isolated phagophore, often between mitochondria and endoplasmic reticulum [60]. During this stage, ULK1 complex triggers the vesicle nucleation of isolation membrane by phosphorylating components of class III PI3K complex (Beclin 1, autophagy, and beclin 1 regulator 1 (AMBRA1), lipid kinase vacuolar protein sorting (VPS)34, VPS15, UV radiation resistance-associated gene protein (UVRAG), ATG14), which in turn activates local phosphatidylinositol-3-phosphate (PI3P) production at a characteristic ER structure called omegasome [121,123]. Class III PI3K complex also facilitates the localization of autophagic proteins to phagophore. BCL-2 and BCL extra-large (BCL-XL) interact with Beclin 1 in Class III PI3K complex to decrease its pro-autophagic activity [121].

In the following stage, the growing double-membrane undergoes vesicle elongation and is eventually sealed to form a double-layered vesicle called autophagosome. Several cellular membranes, including plasma membrane, mitochondria membrane, recycling endosomes, and, Golgi apparatus, may contribute to the elongation of autophagosomal membrane by donating their membrane material [60,123]. This process is mediated by two ubiquitin-like (UBL) conjugation systems [121]. One involves the conjugation of phosphatidylethanolamine (PE) to cytoplasmic protein light chain 3 (LC3)I to generate a lipidated form LC3II, which is facilitated by protease ATG4B and ATG7. ATG12-ATG5-ATG16L1 complex is also required for the formation of covalent bond between LC3 and PE [122]. LC3II will be incorporated into the growing membrane of autophagosome [121]. Another is mediated by ATG7 and ATG10, resulting in an ATG5-ATG12 conjugate. Subsequently, several soluble NSF attachment proteins receptor (SNARE)-like proteins such as syntaxin 17 (STX17), mediate the fusion between autophagosomes and lysosomes and ultimately form autophagolysosomes [60,121]. LC3II as the characteristic signature of autophagic membranes remains associated with autophagosomes and autolysosomes, facilitating their identification [60,123]. In the selective autophagy process, LC3II is critical for the sequestration of specifically labeled cargo into autophagosomes [123]. Most assays for autophagy evaluate the redistribution of LC3II to autophagosomes and autolysosomes by immunohistochemical labeling or by fluorescent imaging in cells after fusion to fluorescent proteins, such as green fluorescent protein (GFP) [60]. During the degradation stage, many lysosomal enzymes, such as acidic hydrolases, can degrade and hydrolyze the cargo in autophagolysosomes, such as proteins, nucleic acids, and lipids at a low optimum pH14 [60,121]. The salvaged nutrients are released back to cytoplasm to be recycled by cell [123].

Autophagy in the CAP-treated cancer cells has been demonstrated in recent studies. The oxidative stress due to CAP treatment has been regarded as the main reason to trigger autophagy in several cell lines, including melanoma G-361, endometrial cancer cells (AMEC, HEC50), and mesothelioma cells EM2 established from rats injected with asbestos [104,125,126]. For melanoma cells G-361, the synergistic use of CAP and silymarin nanoemulsion (SN) triggers autophagy by activating PI3K/mTOR and epidermal growth factor receptor (EGFR) pathways [125]. For endometrial cancer cells AMEC and HEC50, mTOR pathway has been inactivated by CAP-treated medium containing long-lived reactive species [126]. The autophagy inhibitor MHY1485 could partially inhibit autophagic cell death in this case [126].

As a representative example, here, we examine the autophagic response of malignant mesothelioma cells to CAP treatment. The autophagic response of malignant mesothelioma cells to CAP was induced by oxidative stress through a reaction involving cellular iron (Figure 4a). The CAP-triggered oxidative stress stimulates the increased fluid-phase endocytosis, lysosomal biogenesis, and autophagy [104]. CAP treatment also caused a rapid nuclear translocation of a transcription factor, transcription factor EB (TFEB), which regulated lysosomal biogenesis and autophagy [104]. The direct transmission electronic microscopic (TEM) observation of autophagosomes (AP) and autophagolysosomes (AL) in the CAP-treated mesothelioma cells are presented in Figure 4b. Fluorescent imaging demonstrated that noticeable autophagosomes (labeled by LC3II) were formed at 2 h after treatment (Figure 4c). Autophagolysosomes were formed at 4 h after treatment, labeled by the colocalization of LC3II and lysosomal-associated membrane protein 1 (LAMP1).

## 4. Necrosis

Necrosis is essentially different from typical programmed cell death, such as apoptosis. Necrosis is characterized by organelle swelling or the cytoplasmic membrane rupture with the spillage of intracellular contents [115]. Necrosis can be caused by diverse reasons, including metabolic failure, with rapid depletion of adenosine triphosphate (ATP), acute hypoxic, ischemic injury, and trauma, such as mechanical damage on cellular membrane [115,127]. Such a strong leak of cellular components into extracellular space will trigger inflammatory response in vivo [128,129]. Most necrotic processes do not have clear biochemical cascade pathways [129].

Necrosis in the CAP-treated cancer cells has been mentioned many times in literatures mainly based on flow cytometric data. The direct observation of necrosis has not been reported until very recently. Using a novel experimental design, it was found that melanoma cells B16F10 were effectively killed by physical factors mainly the EM effect from a typical CAP jet source [20]. Similar phenomena have also been observed later during the treatment on glioblastoma cells U87MG, lung cancer cells A549, bladder cancer cells MBT2, MB49, and breast carcinoma cells MDA-MB-231 [130,131]. In these cases, CAP treatment was performed on the bottom of an inverted cell culture dish or a multi-well plate. This novel, experimental design-enabled CAP just exerted physical effects on cells without exerting any chemical effect (Figure 5a). Besides, even plasma did not contact the target; necrosis still occurred when there was an air gap around 8 mm [20]. The EM effect from CAP should cause such necrosis.

The physically based CAP treatment caused such a necrosis, characterized by cytosol aggregation and bubbling on cellular membrane (Figure 5b). The cytosol aggregation occurs nearly immediately (<1 min) after a short treatment lasting 2 min. If the treatment lasts longer than that, cellular changes may already start before the end of treatment. The cytosol aggregation is a very fast process, lasting probably only 1 min or even less. After that, the cellular shape does not experience any noticeable change for days except bubbling immediately after cytosol aggregation. The fast bubbling on cytoplasmic membrane of the aggregated cells may be triggered by cytosol aggregation and facilitated by potential holes on membrane based on the fluorescent observation of new small bubbles’ growth process [20]. The whole growth of bubbles only lasted about 8–11 min (Figure 5b). Over the following 2 h, there was no obvious change in the bubbles’ size until the final detachment of bubbles from cells. After this stage, cells did not show any cellular activities, such as division and mobility. Fluorescent imaging shows that there might be no organelle in bubbles either attached or detached from cellular membrane (Figure 5c). Bubbling should be due to the leak of cellular solutions from cytoplasm membrane. It is reasonable to speculate that there is a plasma membrane as the interface between the solutions in bubbles and the extracellular environment.

The CAP-triggered change on the intracellular osmotic pressure may trigger bubbling. Though the mechanism is unknown, the strong aggregation of cytosol may be the direct factor to push the cellular solution out of cytosol membrane. Hypotonic environment, such as deionized water exerted a strong extracellular osmotic pressure. Both B16F10 cells and U87MG cells experienced noticeable swell and final burst in it. In contrast, when the physically based CAP-treated B16F10 cells and U87MG cells immediately cultured in deionized water, they indeed experienced clear swell but without finally losing the integrity of cytoplasmic membrane [20,130]. Thus, the mechanical property of cytoplasmic membrane may be enhanced and can even resist the strong osmotic pressure. Besides, bubbling could also be largely inhibited when the physically based CAP-treated cells were immediately moved to hypotonic solutions, such as deionized water [20,130].

Furthermore, the necrosis due to physical factors is a much faster process than typical apoptosis and autophagy. Cytosol aggregation and bubbling are typical features in the initial stage of this necrosis, which totally lasts only about 10 min. The detachment of bubbling from cellular membrane finishes about 2 h from CAP treatment. In contrast, a typical apoptotic process, from the initial death stimulus to the final formation of apoptotic body, usually lasts at least several hours [132,133]. Like apoptosis, autophagy also occurs relatively slowly in cancer cells. As an example, a time-lapse observation found the clear formation of autophagosome in H4 neuroglioma cells started around 1 h after the stimulus [133]. The slow development of apoptosis and autophagy is due to the cascade of biochemical pathways underlying the programmed cell death. Authors proposed that such a necrosis may not involve these programmed death pathways but shows just a quick response to the physical factors in CAP [20,130].

## 5. Bacterial Cell Death

Bacteria are typical prokaryotes without typical organelles, such as mitochondria, ER, Golgi apparatus, nucleus, peroxisome, cytoskeleton, and lysosome. Compared with mammalian cells with a single layer cytoplasmic membrane, bacteria have a unique physical barrier comprised of capsule, cell wall, cell envelope, and cytoplasmic membrane. Bacteria can be divided into two groupings based on their responses to Gram stain: gram-positive and gram-negative. Gram-positive bacteria will retain crystal violet stain after Gram staining and vice versa for gram-negative cells. Such a different response is due to different compositions and structures of bacterial cell walls. The cell walls of gram-positive bacteria, such as streptococcus and Corynebacterium, comprise a thick layer of peptidoglycan, which contains lipids and other protein components, surrounding a lipid membrane [134]. In contrast, gram-negative bacteria, such as *E. coli* and *P. aeruginosa,* possess a much thinner peptidoglycan layer (cell wall) in periplasm sandwiched between two cell membranes [134,135,136]. Outer membrane contains proteins, such as porins and lipopolysaccharides (LPS). Inner membranes contain various proteins, such as many transporters for metabolites.

Bacteria are the pathogens of many human diseases, particularly dermatitis and wound infection [137]. In addition, bacteria cause serious contamination and deterioration, which is a large threat to food and crop industry [138]. Among them, the formation of biofilms is a common bacterial safety concern. Microbial biofilms are populations of microorganisms that are concentrated at an interface and typically surrounded by an extracellular polymeric substance (EPS) matrix [139]. In fact, most of the Earth’s prokaryotes live in biofilms as organized communities rather than as a unicellular unit [140]. The bacterial collective interaction in biofilm relies on quorum-sensing systems [141,142]. Due to complex interactions and communication in biofilm, killing bacteria in a biofilm is much more difficult than killing these cells in a single cellular status [139].

CAP is an effective sterilization tool to kill bacteria and inactivate biofilm [32,34]. Killing bacteria is also a key factor to improve the wound-healing efficacy [143,144]. CAP also can inactivate many food-borne pathogens [145,146]. So far, most studies focused to describe the inhibition capacity of CAP on bacteria in vitro [36,147,148]. The direct microscopic observation of such cellular death is rare. These studies tend to explore the underlying molecular pathways or mechanisms are even rarer [30,149,150]. The CAP-killed bacteria shows noticeable structural damage on cell wall and complete disassembly of a whole bacteria [151].

One representative study is shown in Figure 6. The CAP-caused bacterial death of two gram-negative bacteria (*E. coli* and *P. aeruginosa*) and two gram-positive bacteria (*S. aureus* and *B. subtilis*) were studied when these bacteria were cultured on an agar plate. It is found that different scenarios of CAP-induced bacterial death could be triggered depending on discharge voltages, treatment time, and bacterial strain (Figure 6a). A short CAP treatment caused ROS stress in bacteria and triggered a programmed cell death in bacteria with several hallmarks of apoptosis. When CAP treatment was sufficiently long, the cell death of bacteria would be a completely physical destruction characterized by the leak of intracellular components from cell wall and a complete collapse of cellular structures [149]. Similar features have been observed in many cases [30,34,149,150,151,152,153]. The increased discharge voltage resulted in a stronger physical damage (Figure 6b).

The thicknesses and physical structure of gram-positive and gram-negative bacteria directly determines different mechanical properties of bacterial cell walls. Gram-negative bacteria are more susceptible to the physical/mechanical stress-triggered disintegration than gram-positive bacteria [149,154]. In CAP treatment, the inverse proportional correlation between the cell wall thickness and the killing rate has been demonstrated. Gram-negative bacteria were more susceptible to the CAP-induced physical destruction than gram-positive bacteria [149]. In another study, it is found that the biofilms of gram-negative bacteria were inactivated more rapidly than the biofilms of gram-positive bacteria [31]. Furthermore, the components in biofilm, such as extracellular matrix (ECM), cell membrane, proteins, and DNA, may also affect the inactivation efficacy [31]. Compared to the biofilm with mono-bacterial species, the biofilms with multi-bacterial species also showed a quite different response to CAP [31]. In short, the synergistic nature of reactive species components and physical factors in CAP make it a powerful tool to inactivate biofilm through either the chemically-triggered cell death or the direct physical destruction on bacterial structure.

## 6. Viral Inactivation

There is no such a concept of viral apoptosis or necrosis because viruses are not cells. CAP has been regarded as a novel, promising anti-viral tool in terms of sterilization on different substrates. Both chemical factors, such as reactive species and physical factors, may play some roles in such an inactivation process by either chemical modification on virus capsid or by direct structural damage on virus. Despite the absence of basic cellular structures, like cell walls, a virus has its own capsid composed of protein and phospholipid membrane. Here, we representatively introduce two examples as the demonstration.

CAP can inactivate bacteriophage, a typical bacterial virus [155,156]. One typical study on different bacteriophages is examined below. The inactivation efficacy of direct CAP treatment and indirect CAP treatment based on the CAP-treated water has been compared on three bacteriophages with different genetic materials: T4 (double-stranded DNA), ϕ174 (single-stranded DNA), and MS2 (RNA) [156]. T4 bacteriophages showed a stronger resistance to a long-term CAP treatment than the other two types. Direct CAP treatment showed a much stronger inactivation efficacy compared to the CAP-treated water treatment [156]. Clearly, such a difference may be due to short-lived reactive species and the physical factors in CAP (Figure 7a). TEM-negative stained imaging of the treated bacteriophages revealed the aggregation of bacteriophages after treatment (Figure 7b). The DNA in T4 bacteriophages might cross-link with themselves or with viral capsids [156]. Though reactive species have been regarded as the main inactivation factors, the aqueous layer covering the virus may be a natural blocker for physical factors. Thus, it is still questionable whether physical factors alone can also cause similar or even stronger structural damage on the capsid.

CAP also shows strong inactivation capacity on mammalian viruses, such as adenoviruses, immunodeficiency virus, and calicivirus [48,157,158]. Here we introduce the study on a feline calicivirus (FCV) as a representative example of mammalian virus. FCV has a globular shape capsid and uses RNA as its genetic material. A cold argon-oxygen plasma source oxidized and disintegrated the capsid protein of a feline calicivirus [48]. CAP treatment inactivated FCV mainly by damaging the viral capsid by oxidizing specific amino acid residues located in the shell (S), the protrusion (P) domains, as well as the dimetric interface regions of major capsid protein in FCV (Figure 8a). These chemical changes on the capsid might cause the loss of virus structural integrity and even the distortion of FCV virions (Figure 8b). A short-term treatment inactivated FCV by oxidizing specific functional peptide residues located in specific domains responsible for the virus attachment and entry to the host cells [48]. The oxidative stress also oxidized and damaged viral RNA when the capsid collapsed and exposed RNA (Figure 8c,d).

Over the last year, severe acute respiratory syndrome coronavirus 2 (SARS-CoV-2) has been getting extreme attention world-wide as a cause of the global COVID-19 pandemic. For coronavirus, spike proteins, like S protein in SARS-CoV-2 and SARS-CoV (2002 SARS virus), play the key role to trigger the initial receptor binding and internalization of virus particles into host cells [159,160] (Figure 9a). The structural biology studies revealed the details between two binding domains at an atomic level resolution [161,162] (Figure 9b). Take SARS-CoV-2 as an example: 8 amino acids residues, N487, Y489, Y453, Y449, G496, T500, G502, and Y505, on the receptor-binding domain (RBD) of S protein interact with corresponding amino acids residue at the binding interface of human receptor angiotensin-converting enzyme 2 (ACE2) [161].

The drugs or methods to affect or to inhibit the binding of S protein with ACE2 will have potential therapeutic values to stop the transmission of such a virus [163]. The copious reactive chemical environment and composition of CAP may cause damage or chemical modification on these key amino acid residues at the binding interfaces and further inactivate SARS-CoV-2. CAP can react nearly all 20 amino acids, mainly targeting their residues [164]. Among 20 amino acids, CAP is very reactive with cysteine (C), methionine (M), tryptophan (W), phenylalanine (F), and tyrosine (Y) [72,164,165]. CAP treatment can add more hydroxyl groups (-OH) and nitro groups (-NO_2_) on the aromatic ring of tyrosine by hydroxylation and nitration, which may directly affect the recognition between RBD and ACE2 through hydrogen bonds [164,165]. This speculation is also supported by the demonstration of CAP inactivation on FCV [48]. Thus, CAP may be a powerful tool to inactivate SARS-CoV-2 and may have wide use as a sterilization tool to fight the COVID-19 pandemic.

## 7. Cross-Species Similarities

Several general trends or similarities of the CAP-caused biological destruction can be summarized here, which have been widely observed in mammalian cells, bacteria, and viruses. First of all, direct CAP treatment causes stronger damage to cells and viruses than the indirect CAP treatment based on CAP-treated solutions, such as medium, PBS, and water [148,156,166,167]. Direct CAP treatment involves the impact of long-lived reactive species, short-lived reactive species, and physical factors. In contrast, indirect CAP treatment will just cause the biological effect due to long-lived reactive species. For cancer cells, long-lived reactive species are the key to cause cytotoxicity [72,73]. However, long-lived reactive species alone cannot explain the observed anti-cancer efficacy. Short-lived reactive species may contribute to some unique cellular responses of cancer cells, making the stronger cytotoxicity of direct treatment than indirect treatment. On one hand, the strong micromolecular cells-based H_2_O_2_ generation has been observed in directly CAP-treated cancer cells [168,169,170]. On the other hand, directly CAP-treated cancer cells enter the activation state and become very sensitive to the cytotoxicity of long-lived reactive species like H_2_O_2_ and NO_2_^-^ [171,172]. These two cellular responses do not appear in indirect CAP treatment [173]. Whether bacteria also take part in these two cellular responses to CAP treatment is still unknown and requires dedicated studies.

Second, physical factors may play a key role to trigger necrosis or structural damage on cells and viruses. In previous studies on mammalian cells, the biological effects from physical factors were not validated experimentally. The cell death of the physically based melanoma cells and glioblastoma cells first demonstrated that necrosis could be caused by the EM effect from CAP [20,130]. Based on this conclusion, we speculate that a clear, structural damage on bacteria and virus may be at least partially due to the same physical factors in CAP (Figure 10). For cancer cells, the switch to change the dominant role of physical factors or chemical factors is the medium layer above or surrounding cells [20,130]. When such a layer is adequately thin, physical factors may play the dominant role. If all physical factors have been blocked or absorbed by such as a medium layer, reactive species will dominate the biological effect of CAP. We emphasize that the culture method for mammalian cells and bacteria were essentially different in most cases. For bacteria, the solid medium, such as specific agar, has been widely used in many cases. The bacteria grow above the solid medium layer and get nutrients from it. When a CAP treatment is performed in such a case, there is no thick medium layer to cover these bacteria. In these cases, physical factors may be the dominant factor. Whether this trend also exists in the CAP-treated virus is still unknown and requires further studies.

Among three physical factors, the near-room-temperature nature of CAP and UV radiation are not likely to cause observable physical destruction in both mammalian cells and bacteria. For cancer cells, such as melanoma cells, the heating treatment with the same temperature of the CAP tip did not cause noticeable cellular changes [20]. The blockage of UV irradiation from CAP by a filter layer did not change the physically triggered necrosis in melanoma cells either [20]. UV in CAP also did not kill *S. Typhimurium* cells grown on a solid medium [150]. The EM effect from CAP may be mainly attributed to the necrosis observed in six cancer cell lines [20,130]. Besides, for reactive species-triggered apoptosis, the rise of intracellular ROS is the main mechanism to trigger the final cell death. The pretreatment of intracellular ROS scavengers, such as NAC, can effectively protect cells from the cytotoxicity of CAP treatment [76,77,78,79,80,81,82]. However, the pretreatment of NAC in bacteria cannot inhibit the physically triggered structural damage [149]. Our unpublished data also demonstrate that the pretreatment of NAC does not stop the physically-triggered necrosis in the CAP-treated melanoma cells. These results further confirm that physical factors kill cells using essentially different pathways compared with reactive species.

## 8. Conclusions

CAP has shown promising applications in many branches of medicine, including bacterial sterilization, wound healing, virus inactivation, and cancer therapy. The destruction of biological targets is a foundation to understand the mechanisms of plasma effects on biological objects. Both chemical factors, such as reactive species, and physical factors, such as the EM effect, can cause biological destruction in specific forms. When reactive species dominate biological effects, apoptosis tends to be the main cell-death type. When physical factors dominate, necrosis or direct structural damage on cells may be the main cell-death type. For mammalian cells, when treatment is directly performed on cells, due to the coverage of medium, apoptosis is the main cell-death mechanism. When cells can be directly exposed to physical factors without the interference of aqueous layer, physical factors tend to cause damage on cellular structure, particularly for the bacteria cultured on solid medium. The multimodality of the CAP-based biological effects that cause diverse cellular responses, particularly biological destruction, after CAP treatment is one of the most unique and attractive features of plasma medicine.

## Figures and Tables

**Figure 1 biomedicines-09-01259-f001:**
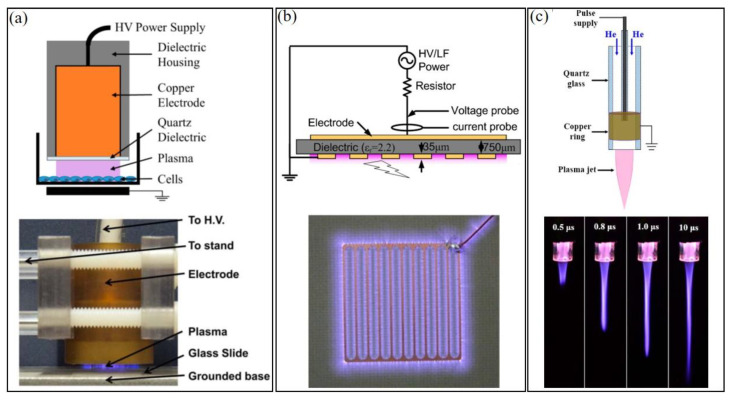
Typical CAP sources. (**a**) A volume DBD source [16,17]. (**b**) A surface DBD source [18]. (**c**) A CAP jet source. The length of plasma jet can be modulated by controlling signals, such as pulse width (µs) [14].

**Figure 3 biomedicines-09-01259-f003:**
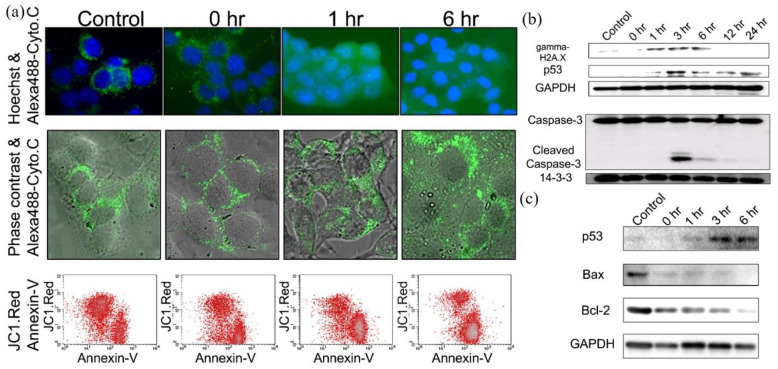
A typical observation of apoptosis in the CAP-treated melanoma cells B16F10. (**a**) Cellular images of cytochrome c release by immunocytochemistry and fluorescence-activated cell sorting analysis to check mitochondria membrane potential change. (**b**) Western blot analysis of the phosphorylation on serine 139 on H2AX histone (γ-H2AX), p53, and caspase-3. (**c**) Western blot analysis of BAX and BCL-2 [18].

**Figure 4 biomedicines-09-01259-f004:**
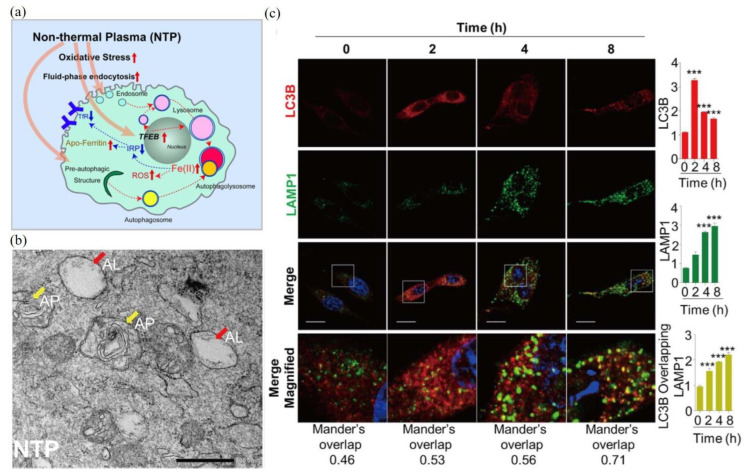
CAP induces oxidative stress response in malignant mesothelioma cells (SM2), resulting in increased endocytosis, lysosome biogenesis, and autophagy-associated cell death. (**a**) Schematic autophagy mechanism of the CAP-treated malignant mesothelioma cells. (**b**) TEM imaging of vesicle structures, endosome-like vesicles, and organelles involved in the autophagic pathway of the CAP-treated SM2 cells (60 s). Yellow and red arrowhead indicates AP and AL, respectively. Scale bar = 500 nm. (**c**) Fluorescent imaging of the formation of autophagosomes in the CAP-treated (60 s) malignant mesothelioma cells. Cells were stained by DAPI (DNA, blue), LC3B (red), and LAMP1 (green). LC3B suggested the presence of autophagosomes, while the colocalization of LC3 and LAMP1 suggests the autophagolysosome formation (yellow overlay fluorescence). Scale bar = 20 μm. ***, *p* < 0.001 [104].

**Figure 5 biomedicines-09-01259-f005:**
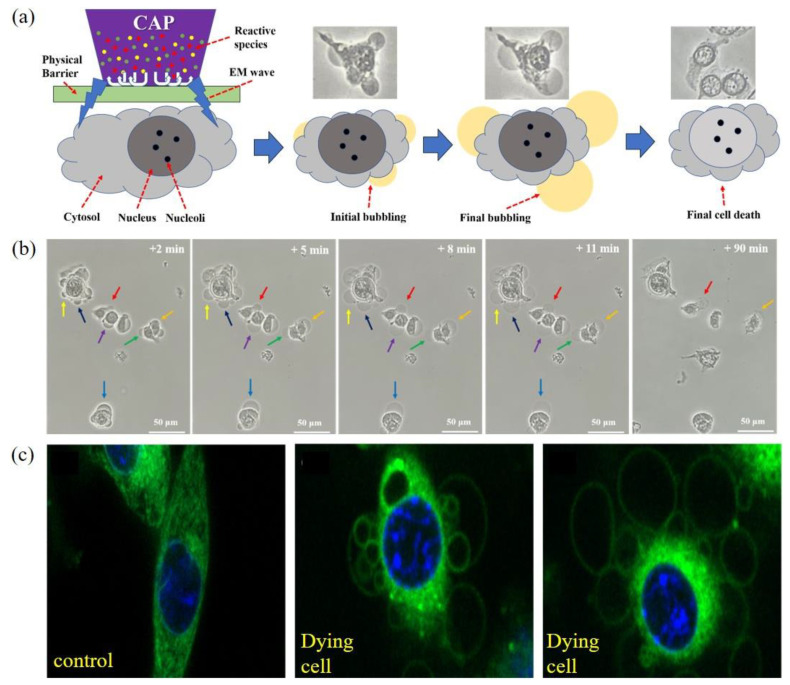
Direct observation of necrosis on the physically based CAP-treated melanoma cells (B16F10). (**a**) Schematic illustration. (**b**) A time-lapse observation of bubbling. (**c**) The fluorescent imaging of the treated B16F10 cells. Microtubules (green) and DNA (blue) were stained using BioTracker 488 green microtubule cytoskeleton dye and Hoechst 33342, respectively [20].

**Figure 6 biomedicines-09-01259-f006:**
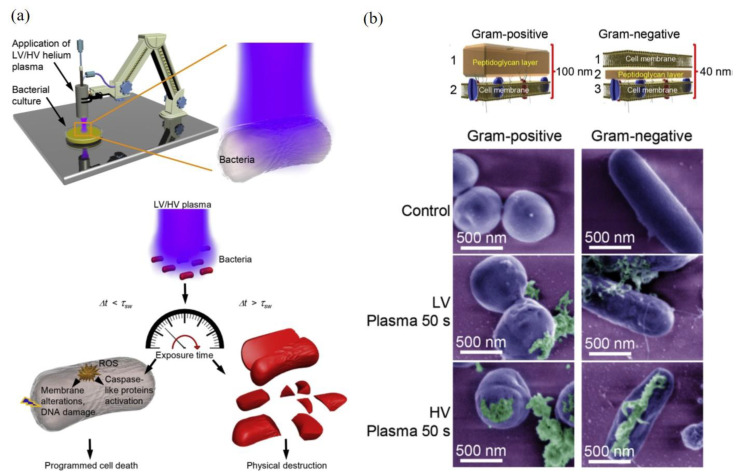
The cell death of bacteria after CAP treatment. (**a**) The interplay of physical destruction and biological cell death upon CAP treatment. (**b**) False-colored SEM images of bacteria exposed to CAP with high discharge voltage (HV) and low discharge voltage (LV) [149].

**Figure 7 biomedicines-09-01259-f007:**
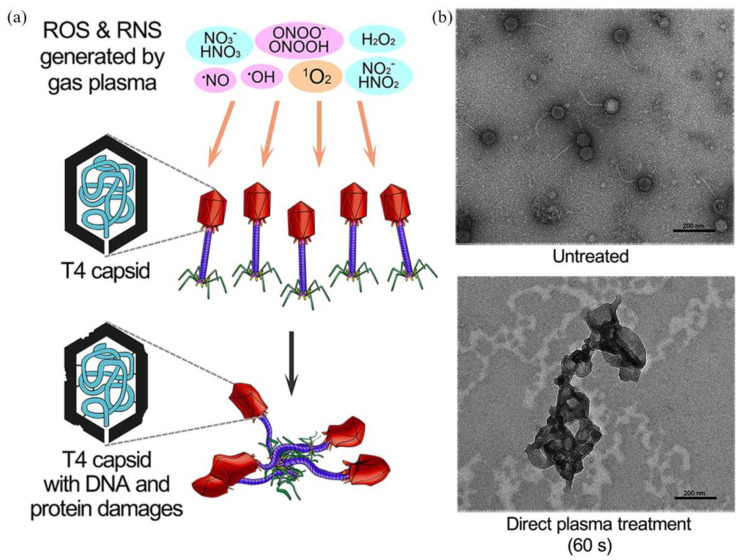
The CAP-triggered damage on T4 bacteriophage. (**a**) Schematic illustration. (**b**) TEM-negative stained imaging of T4 bacteriophage treated by CAP and the CAP-treated water [156].

**Figure 8 biomedicines-09-01259-f008:**
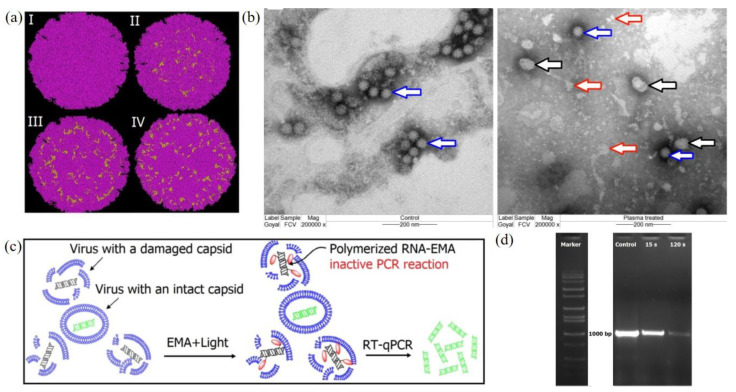
The CAP-triggered damage on FCV. (**a**) TEM imaging of calicivirus the control (**left**) and the experimental group (15s, **right**). Blue arrows, black arrows, and red arrows refer to untreated virus particles, distorted viral particles, and the debris of the damaged viral particles, respectively. (**b**) The calculated location of oxidized peptide residues (yellow colored) among the N-terminal arm (NTA) domain (I); shell (S) domain (II); P1 subdomain (III); and P2 subdomain (IV). (**c**) The principles of the quantification of capsid destruction due to CAP treatment. Reverse transcription PCR (RT-PCR) was used to quantify the unaffected calicivirus’s RNA. (**d**) The agarose gel (DNA) patterns of EMA-coupled RT-PCR products from viral RNA obtained from control, 15 s, and 120 s of CAP treatment [48].

**Figure 9 biomedicines-09-01259-f009:**
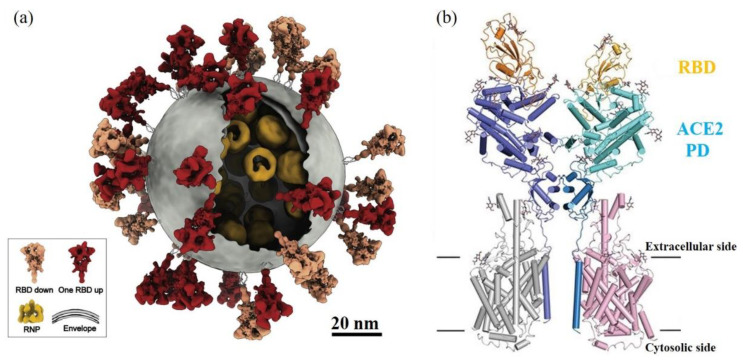
SARS-CoV-2 structure. (**a**) Molecular architecture of SARS-CoV-2 virus obtained by cryoelectron tomography and subtomogram averaging. The S protein in “RBD down” conformation, “one RBD up” conformation, lipid envelope, and ribonucleoproteins (RNPs) are shown in salmon, red, gray, and yellow, respectively [160]. (**b**) The cryoelectron microscopy structure of RBD-ACE2 complex [162].

**Figure 10 biomedicines-09-01259-f010:**
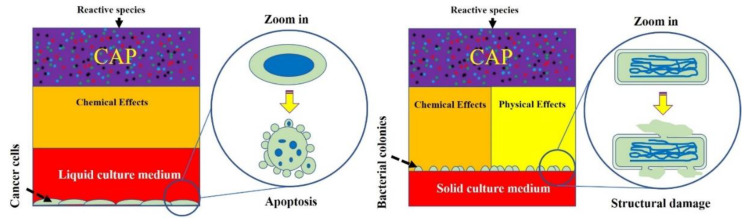
A schematic illustration of typical CAP treatment on cancer cells and bacteria in vitro. In most experimental setups in vitro, cancer cells or mammalian cells were immersed in a layer of medium during direct CAP treatment. In contrast, many bacterial cells were directly exposed to CAP because solid culture medium was widely used in many cases. Such a different experimental tradition may naturally filter the physical effectors of CAP in the studies involving liquid culture medium.

## Data Availability

Not applicable.

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
