# Peer review of "Multi-Modal Biological Destruction by Cold Atmospheric Plasma: Capability and Mechanism"

_biomedicines, 2021, doi:10.3390/biomedicines9091259_

Round 1

Reviewer 1 Report

The authors well described the research issues by the low-temperature plasmas, using cold atmospheric pressure plasmas for biomedical applications. The authors comprehensively treated many interesting issues with lots of citations on the research activities as a good review. However, as a referee of a scientific journal, I feel that the manuscript is just an introduction of a well-written review paper without new scientific results presented by the authors themselves. Therefore, the manuscript itself is not recommended for publication as it is. The authors should mention what are newly added results in this manuscript compared with all the previous researches mentioned here. Please add more description following the comments below.

  1. The title, "Multi-modal biological destruction", is not well described in the main. What is the definition of multi-mode? What is the synergetic effect of different modes?  
  2. It would be better to rewrite the manuscript following the conventional structure with introduction, methodology, results, and conclusion. In this paper, most of the parts belong to only introduction. The authors should include something added solely by their own research.
  3. The abstract also describes introductory explanations only. No main result is explained here. 
  4. The conclusion is quite simple. The description for the variation of the amount of reactive species and physical factors should have been mentioned in the text with the variation of control parameters for the three different types of plasma devices shown in Fig. 1. The final sentence, "The multimodality of the CAP-based biological effects which cause diverse cellular responses particularly biological destruction after CAP treatment, is one of the most unique and attractive features of plasma medicine," should be investigated more deeply. What is the change in the effect with the variation of the treatment time, applied power, gas flow rates, etc.? The explanation is charming but practically does not give a good advance in science.

Author Response

Please read the attachment. 

Reviewer 2 Report

Review Report
The review article with the title of Multi-modal Biological Destruction by Cold Atmospheric
Plasma: Capacity and Mechanism, written by D. Yan et al.
Line 28: please remove the second “ionized” to be as follows:
“CAP is a near room temperature ionized gas composed of products including neutral particles such as
neutral atoms and molecules and charged particles such as ions, electrons, and … “
Line 32-35: Regarding “Non-equilibrium of discharge”, the terminology of non-equilibrium is related to
the physical properties of the CAP. It is not related to the discharge. The CAP is a non-equilibrium plasma.
The CAP is generated by a discharge. You can explain about the discharge much more. Which type of
discharge can generate CAP.
Line 36: “… thermal irradiation, UV irradiation, and EM emissions … “
I understand several manuscripts mentioned the same. But in the physical viewpoint, thermal and UV
radiations are a type of EM. Specifically, when you talk about EM, which range of EM do you mean?
The same problem can be observed in the abstract, in Line 558, also in the Conclusion section, Line 573.
Line 41-43: gas flow is not a discharge parameter. It is a mechanical parameter.
Regarding discharge parameters, in addition to the voltage, you can write about frequency. In the case of
pulsed discharge, you can write about the duty cycle.
In the previous part, first you explain the discharges which can generate CAP as non-thermal plasma. Then
you write about the discharge parameters like voltage, frequency, … . You can explain more about the
types of electrical discharges.
Line 51: Regarding “Bulk CAP”, do you mean plasma jet? You try to make a correlation between a plasma
jet and bulk CAP to avoid confusion. Or if it is different, please explain more. 

Author Response

Please read the attachment. 

Round 2

Reviewer 2 Report

Dear authors,

Thank you for your reply. The authors replied to all my comments and questions.